# Circulating Tumor Cells as Predictive and Prognostic Biomarkers in Solid Tumors

**DOI:** 10.3390/cells12222590

**Published:** 2023-11-08

**Authors:** Maurizio Capuozzo, Francesco Ferrara, Mariachiara Santorsola, Andrea Zovi, Alessandro Ottaiano

**Affiliations:** 1Coordinamento Farmaceutico, ASL-Naples-3, 80056 Ercolano, Italy; m.capuozzo@aslnapoli3sud.it; 2Istituto Nazionale Tumori di Napoli, IRCCS “G. Pascale”, Via M. Semmola, 80131 Naples, Italy; mariachiara.santorsola@istitutotumori.na.it; 3Ministry of Health, Viale Giorgio Ribotta 5, 00144 Rome, Italy; zovi.andrea@gmail.com

**Keywords:** circulating tumor cells, cancer genetics, prognosis, biomarkers, tumor heterogeneity

## Abstract

Circulating tumor cells (CTCs) have emerged as pivotal biomarkers with significant predictive and prognostic implications in solid tumors. Their presence in peripheral blood offers a non-invasive window into the dynamic landscape of cancer progression and treatment response. This narrative literature review synthesizes the current state of knowledge surrounding the multifaceted role of CTCs in predicting clinical outcomes and informing prognosis across a spectrum of solid tumor malignancies. This review delves into the evolving landscape of CTC-based research, emphasizing their potential as early indicators of disease recurrence, metastatic potential, and therapeutic resistance. Moreover, we have underscored the dynamic nature of CTCs and their implications for personalized medicine. A descriptive and critical analysis of CTC detection methodologies, their clinical relevance, and their associated challenges is also presented, with a focus on recent advancements and emerging technologies. Furthermore, we examine the integration of CTC-based liquid biopsies into clinical practice, highlighting their role in guiding treatment decisions, monitoring treatment efficacy, and facilitating precision oncology. This review highlights the transformative impact of CTCs as predictive and prognostic biomarkers in the management of solid tumors by promoting a deeper understanding of the clinical relevance of CTCs and their role in advancing the field of oncology.

## 1. Introduction

Cancer has emerged as a prominent cause of death in both the United States and globally. In contrast to other severe ailments, the mortality rate associated with cancer has been steadily decreasing. In the past decade, there has been an overall reduction in cancer-related deaths, with a notable decline of approximately 33% [1]. This decrease is attributed to a myriad of factors, encompassing more effective cancer screening, enhanced diagnostic tools, advancements in surgical and radiological techniques, and the emergence of systemic treatments such as chemotherapy, immunotherapy, and targeted therapies. The key to further diminishing cancer-related mortality in the future lies in early cancer detection, personalized treatment approaches, and the implementation of effective strategies to prevent or mitigate metastasis. The majority of solid tumor diagnoses are typically established through radiographic and physical examinations, and subsequently confirmed through pathological findings obtained via tissue biopsy. 

An innovative alternative for cancer detection is liquid biopsy, which entails the examination of circulating tumor cells (CTCs) and circulating tumor DNA (ctDNA). Remarkably, although CTCs were identified as far back as 1869 and ctDNA in 1948 [2], their practical application has only recently come to fruition. CTCs have high vitality and possess significant metastatic potential. They originate from primary or metastatic tumors of epithelial origin and enter the bloodstream. CTCs represent a crucial component of liquid biopsy, offering a dynamic window into real-time tumor progression monitoring [3,4]. Unlike traditional high-throughput sequencing analysis of tumor tissue, which involves analyzing a mixed sample of millions of cells and reflects the overall genomic characteristics of the cells, CTCs and cancer stem cells (CSCs), along with other low-abundance yet functionally significant cells, often have their genetic material diluted in such analyses. Thanks to the advent of single-cell sequencing technology, the possibility of isolating and characterizing CTCs is poised to become increasingly integral in cancer management [5,6].

In this narrative review, we explore various CTC detection methods and present the current body of evidence supporting the integration of CTC detection into clinical practice. This article delves into the potential applications of CTCs in solid tumors, encompassing cancer screening, diagnostics, treatment guidance, and surveillance. We have conducted a review and synthesis of pertinent studies available on PubMed and Google Scholar (last accessed on 2 September 2023) regarding the role of CTCs in solid tumors, which are anticipated to influence the future landscape of CTC utilization.

## 2. Role of CTCs in Metastatic Pathway

Metastasis, the dissemination of cancer from the primary tumor to distant bodily locations, represents a highly intricate and dynamic process. This multifaceted event unfolds through a sequence of distinct stages, including local invasion, intravasation, circulation survival, extravasation, and the eventual establishment of colonies in remote organs [7]. Central to this process are CTCs, which have detached from the primary tumor and entered the bloodstream, facilitating their journey to distant sites throughout the body. Their presence in peripheral blood has the potential to predict the development of metastasis, associated with poor prognosis, making them a crucial focus in the study of metastatic pathways [8]. Several interesting studies in the literature suggest that probably only a specific subset of tumor cells possess the ability to metastasize, such as those with clonogenic potential and stem cell properties. Conversely, a significant majority of cells with a tumor cell phenotype may not survive in the circulation [9]. CTCs encompass a diverse array of cells displaying a range of characteristics, encompassing epithelial, mesenchymal, and hybrid phenotypes. The heterogeneous nature of CTCs presents considerable complexities in deciphering their exact functions in the metastatic process. Recent investigations have shed light on the dynamic phenotypic transitions that CTCs experience as they traverse the circulatory system, and these transformations can significantly impact their capacity for extravasation and the establishment of colonies in distant organs [10]. To initiate the metastatic process, tumor cells must initially breach the basement membrane, allowing their eventual entry into the bloodstream through a phenomenon known as intravasation (Figure 1).

This transition involves the downregulation of epithelial cadherin and the upregulation of neural cadherin, leading to the disruption of cell-to-cell adhesion and the acquisition of a mesenchymal phenotype. The degradation of the extracellular matrix is primarily driven by the action of matrix metalloproteinases and the urokinase plasminogen activator system, with their involvement often linked to the process of epithelial–mesenchymal transition (EMT) [11]. In the context of CTCs, there is noteworthy upregulation of EMT-related markers. This upregulation suggests a spectrum in the development of CTC phenotypes, ranging from an epithelial to a mesenchymal differentiation state. Intriguingly, various cancer types exhibit distinctive CTC phenotypes [12]. For instance, in lobular tumors, CTCs predominantly display epithelial characteristics, whereas in triple-negative and human epidermal growth factor receptor 2 (HER2)-positive tumors, CTCs tend to adopt a more mesenchymal profile [13]. Furthermore, in patients responding to therapy, mesenchymal CTCs were found in reduced numbers. In an intriguing study, the role of transforming growth factor-beta (TGF-beta) in EMT has been proposed, with EMT-related markers demonstrating more accurate prognostic value compared to epithelial markers. Therefore, the inclusion of EMT markers is believed to aid in identifying the more aggressive CTC phenotype, as these cells possess the potential for extravasation and adaptation to the microenvironment [14]. 

One of the critical challenges CTCs face is surviving in the hostile environment of the bloodstream. Platelets play a pivotal role in protecting CTCs from immune surveillance, thereby enhancing their survival. Understanding the interactions between CTCs and platelets is essential for unraveling the mechanisms that enable CTCs to evade immune defenses. Indeed, it has been discovered that when entering the systemic circulation, CTCs encounter a hostile environment. Platelets aggregate with these tumor cells, shielding them from natural killer cell-mediated lysis [15]. Consequently, a higher platelet count is associated with reduced survival. Conversely, treatment with anticoagulants has shown the ability to decrease metastasis [16,17,18]. The microenvironment at the metastatic site plays a pivotal role in determining whether CTCs can successfully colonize and establish secondary tumors. This microenvironment comprises various cell types, including endothelial cells, stromal fibroblasts, and immune cells. Crosstalk between CTCs and the microenvironment can modulate the fate of disseminated tumor cells and influence their dormancy or proliferation. For example, it has been discovered that tumor-associated macrophages enhance invasiveness through the release of epidermal growth factors [19]. A fascinating aspect of CTC biology is their ability to enter a state of dormancy, where they remain quiescent for extended periods. This dormancy period can be influenced by various factors, including immune surveillance, angiogenic signals, and interactions with the microenvironment. Understanding the mechanisms underlying CTC dormancy and reactivation is essential for developing targeted therapeutic strategies [20]. 

Certain malignancies exhibit a predilection for particular organs during the metastatic process, a phenomenon that is orchestrated by chemokines such as stromal cell-derived factor-1 (SDF-1) and CCL21. These chemokines are typically expressed at the sites of metastasis, and tumor cells frequently exhibit an overexpression of the corresponding receptor, CXCR4 [21]. To facilitate the reception of incoming tumor cells, the microenvironment at the metastatic site needs to be prepped accordingly. Some research indicates that hematopoietic progenitor cells play a role in creating this conducive niche. When tumor cells interact with the local microenvironment, they undergo a transition to an epithelial phenotype, enabling adhesion and proliferation. Consequently, secondary metastatic lesions often closely resemble the primary tumor in terms of phenotype. This phenomenon, known as mesenchymal–epithelial transition (MET), is marked by characteristics such as unlimited replicative potential, tissue invasion, and metastatic spread [22]. CTCs play a crucial role in the metastatic process, and their investigation will yield valuable insights into the mechanisms governing cancer spread, thereby creating new opportunities for the development of targeted therapies aimed at disrupting the metastatic cascade and enhancing the outlook for cancer patients.

## 3. Circulating Tumor Cells: Detection and Isolation

CTCs are highly dynamic cancer cells found in the peripheral blood of individuals afflicted with solid tumors. They serve as a crucial component in liquid tumor biopsies, holding significant diagnostic value. Both the quantity and the phenotypic characteristics of CTCs demonstrate a notable correlation with the progression of the primary tumor. The scrutiny and examination of CTC quantities and phenotypes offer an indirect means to glean insights into the characteristics of the tumor lesions [23]. The application of CTCs for monitoring the advancement of solid tumors through peripheral blood analysis has gained widespread recognition. Given the relatively low prevalence of CTCs in peripheral blood in comparison to other blood cell types, distinguishing them poses a considerable challenge. In the context of liquid biopsy, approximately one CTC can be identified among ten million white blood cells per milliliter of blood [24]. Adding a layer of complexity, akin to primary tumors, these circulating cells demonstrate heterogeneity, featuring multiple subpopulations expressing diverse molecular markers [25]. The absence of a universally shared marker or molecular signature across all cancer cells poses a formidable challenge in devising highly accurate CTC isolation techniques. To address this challenge, innovative isolation devices have been developed, focusing on leveraging epithelial marker proteins that are expressed on tumor cells while being absent in surrounding blood cells. As biological technologies continue to advance, especially with the development of novel nanomaterials and microfluidic systems, several techniques have emerged for the detection of peripheral CTCs in blood [26]. These methodologies can be broadly classified into two primary categories: label-dependent (Table 1) [27,28,29,30] and label-independent detection (Table 2) [31,32,33,34,35,36,37]. 

Label-dependent isolation methods rely on the use of antibodies that selectively target cell surface antigens found in circulating tumor cells (CTCs). Unlike regular blood cells, CTCs are characterized by the presence of epithelial markers, such as epithelial cell adhesion molecules (EpCAM) and cytokeratin (CK) [38,39]. Specific techniques that leverage this principle include the conjugation of antibodies to magnetic nanoparticles and the use of microfluidics [40,41]. While there is some variability in the phenotypic and functional definitions of CTCs across different studies, it is widely agreed upon that CTCs typically lack the CD45 marker and exhibit the presence of EpCAM and CK [42]. It is worth noting that the expression pattern of cytokeratin may vary among different types of cancer and often mirrors the pattern observed in tissue biopsy specimens. Furthermore, this pattern can change over time as patients undergo treatment or experience disease progression. Label-independent detection methods allow for the isolation of CTCs without relying on cell surface markers. Instead, these techniques focus on distinguishing CTCs from blood cells based on differences in their size, deformability, and various biophysical characteristics [43]. This approach is centered on discerning the unique properties of CTCs to effectively separate them from regular blood cells. CTCs have been detected in the peripheral blood of patients with different types of solid tumors [44]; indeed, there exists a substantial volume of research validating the diagnostic and prognostic significance of CTC examination in advanced breast [45], prostate [46], liver [47], gastric [48], lung [49], and colorectal [50] cancers. 

As previously elucidated, the majority of cancers originate from epithelial cells, making EpCAM the most commonly used marker for CTCs. EpCAM is considered a “universal” epithelial marker for various cancer types [44,45,46,47,48,49,50]. However, the utility of EpCAM as a CTC marker is constrained. It cannot be applied to tumors that lack EpCAM expression or have low levels of expression, such as neurogenic cancers [51]. Additionally, CTCs can undergo epithelial-to-mesenchymal transition (EMT), during which epithelial markers like EpCAM are downregulated [52]. This phenomenon hampers the detection of EpCAM-positive CTCs. Hence, it is noteworthy that relying solely on the detection of EpCAM-positive CTCs likely underestimates the total CTC population and overlooks crucial biological information pertaining to EpCAM-negative CTCs, i.e., those undergoing EMT. From a methodological perspective, the challenge is compounded by the fact that many EMT-related molecules are cytoplasmic or nuclear proteins, rendering them unsuitable for use with currently available membrane-based molecular technologies for CTC detection. Therefore, single-cell CTC sequencing technologies (see beyond) may prove more beneficial for enhancing our understanding of the phenotype of EpCAM-negative CTCs and improving their isolation, enabling a more comprehensive assessment of the EMT status of CTCs at the RNA level.

In our opinion, despite the limitations posed by the heterogeneity of marker expression patterns, label-dependent techniques employing antibodies to target CTCs are the most efficient and specific means of CTC isolation. It is also important to note that label-dependent methods are widely used and firmly established in the field. In other words, the biological isolation of CTCs is a highly compelling area of study, and its potential will grow as our knowledge of tumor phenotypes continues to expand. Therefore, the current limitations of label-dependent methods for isolation are more closely related to the constraints of our oncological knowledge than to any inherent “flaw” in the method.

## 4. Clinical Utility of CTCs: Emergence as Multifunctional Biomarkers

As the body of evidence on the prognostic significance of CTCs continues to expand, researchers have initiated studies to explore interventions that could enhance the survival outcomes for patients facing a grim prognosis marked by elevated CTC counts or an adverse change in CTC levels. Additionally, investigations have been carried out to ascertain whether the biological insights derived from CTCs hold the potential to advance patient care. Indeed, it is crucial to give significant attention to circulating tumor cells (CTCs), viewing them not merely as progression disease (PD) endpoints, but also recognizing their role as prognostic indicators and as biomarkers for predicting and assessing responses at intermediate endpoints [53]. Various investigations have substantiated the prognostic relevance of CTC counts across diverse tumor types. These studies offer additional confirmation of the tumoral origins of CTCs [54,55,56]. In particular, CTC enumeration serves as a current prognostic marker for progression-free survival (PFS) and overall survival (OS) in breast, colorectal, and prostate cancer [57]. The utilization of CTCs as a surrogate biomarker involves establishing a minimum cutoff value to assess prognostic outcomes. However, this cutoff value varies depending on the isolation device, the protein marker employed, and the origin of the primary tumor. For instance, a retrospective analysis determined that ≥5 CTCs in 7.5 mL of blood signifies a cutoff value indicating poor prognosis for OS and PFS in metastatic breast cancer patients when an EpCAM-dependent device is used (CellSearch System^®^ isolating CTCs by using EpCAM-based capture, fluorescent labeling, and computer-assisted identification, distinguishing them from CD45+ leukocytes) [58]. This determination led to the establishment of “BCTC-positivity” as a widely adopted cutoff value for prognostication in breast cancer [59]. A similar retrospective determination of CTC cutoff values was observed in clinical trials for metastatic prostate cancer, where the use of ≥5 CTCs as a cutoff correlated with OS [60]. In colorectal cancer (CRC), retrospective findings associated >3 CTCs with OS and PFS [61]. Although correlations between enumeration and survival rates have been established [62], determining optimal cutoff values remains a challenge, particularly for cancers positive for epithelial markers, with variations across trials [63,64,65]. Studies focusing on other cancer types encounter difficulties in defining prognostic cutoff values [66], underscoring the need for more effective CTC isolation techniques and retrospective analyses. 

Beyond serving as a prognostic biomarker, CTC enumeration is employed as an exploratory indicator of efficacy and a measure of treatment response [67]. Notably, a decline in CTC counts demonstrated promise as an early efficacy endpoint in a study examining cabazitaxel response in metastatic castration-resistant prostate cancer patients with docetaxel resistance [68]. Enumeration was also evaluated as a patient stratification biomarker in the SWOG S0500 trial, where changes in CTC counts in response to therapy were used as an early indicator of therapy resistance, although this prospective use proved ineffective at extending overall survival in metastatic breast cancer patients [69]. To enhance the utility of CTCs as a biomarker, enumeration is employed as a secondary outcome measure to improve the accuracy of other prognostic markers utilized in clinical practice, such as tumor-associated protein markers (e.g., PSA, CEA, CA125) and imaging tests. A specific study revealed that combining CTC enumeration with positron emission computerized tomography (PET-CT) had greater prognostic significance than either measure alone [70]. Ensuring uniformity in isolation and characterization methods is crucial to explore the predictive value of CTCs in the clinical setting. Without uniformity, there is inconsistency in the analyzed CTC population, encompassing phenotypic variability and inconsistent capture efficiency in various cancer types, thus limiting the scope of significant findings [71]. Moreover, the identification of new markers poses technical challenges [72]. Technological advancements and extensive clinical validation are essential to understand CTCs’ role in guiding personalized medicine and drug development. 

Detecting CTCs holds a critical role in cancer prognosis, influencing clinical decision-making. Prognostic assessment is pivotal for weighing the risks and benefits of proposed treatments. Extensive efforts have been directed towards unravelling the clinical significance of CTCs in prognostic estimation and their role in guiding therapeutic decisions. To illustrate and validate the prognostic, and at times diagnostic, implications of CTCs across diverse cancer types, we have identified some studies utilizing various CTC detection technologies. In most cases, as evidenced by the selected studies, patients with a heightened count of CTCs (deemed unfavorable) faced a poorer prognosis, primarily assessed through PFS and OS estimates (Table 3) [73,74,75,76,77,78,79,80,81]. The outcomes of these investigations not only clarify the tangible potential of CTC enumeration but also offer profound insights into the feasibility of using CTCs as liquid biopsies.

## 5. Characterization of CTCs: Beyond Diagnosis and Prognosis

In the realm of cancer research, the characterization of CTCs transcends conventional diagnostic and prognostic paradigms. Expanding beyond their role as biomarkers for disease presence and outcome prediction, a nuanced understanding of CTCs necessitates an exploration of their molecular and functional attributes. Unraveling the heterogeneity within the CTC population and deciphering the underlying mechanisms governing their dynamics becomes imperative. Advances in single-cell analysis techniques and high-throughput technologies have propelled investigations into the diverse phenotypic and genotypic profiles of CTCs, shedding light on their potential as indicators of therapeutic response and avenues for targeted interventions. Certainly, extending past the realms of diagnostic applications and prognostic guidance, CTCs play a pivotal role as pharmacodynamic biomarkers, gauging both inherent and acquired resistance in response to distinct treatment modalities. The molecular profiling of CTCs has exerted a discernible impact on tailoring therapeutic strategies in specific instances. Moreover, it has advanced our comprehension of the intricate mechanisms governing cancer metastasis, concurrently uncovering novel therapeutic targets amenable to intervention. Several studies in the literature suggest the potential use of CTCs as predictive markers for the selection of administered chemotherapy [82,83,84]. Moreover, studies demonstrate the potential role of CTCs combined with ctDNA as a complementary biomarker for the early indication of treatment response [85,86]. Among these, we would like to emphasize an intriguing study—a phase II clinical trial conducted by Punnoose et al. investigating erlotinib and pertuzumab in patients with advanced non-small-cell lung cancer (NSCLC). In this study, the decrease in CTC count after treatment was correlated with a longer PFS. Interestingly, patients with EGFR mutations exhibited a substantial reduction in CTC count during treatment. Mutational analysis of EGFR revealed that ctDNA had a higher sensitivity in detecting mutations compared to CTCs, and, after treatment, a decrease in mutational burden suggested a partial response to the treatment [87]. 

Similar to CTCs, ctDNA does possess certain limitations. Firstly, ctDNA solely facilitates the genotypic characterization of malignancy, in contrast to CTCs, which permit phenotypic analysis. Furthermore, the sensitivity of ctDNA might exhibit variations based on organ involvement. Notably, investigations into recurrent metastatic CRCs have indicated higher sensitivity in cases of hepatic metastasis compared to pulmonary metastasis [88].

The study of circulating tumor cells (CTCs), including novel single-cell techniques, can offer valuable insights into the realm of personalized therapy. In this context, we can reconcile the early characterization of tumor heterogeneity with the resolving power and informative capacity of emerging single-cell technologies. Briefly, single-cell sequencing stands as a groundbreaking approach that allows for the examination of the genetic material of individual tumor cells. This methodology provides insights into the genome, transcriptome, and other multi-omics data at the single-cell level. It is important to note that this section will not delve into the specifics of single-cell whole-genome amplification techniques (scWGA). For detailed information on certain methods such as MALBAC, eMDA, LIANTI, SISSOR, and META-CS, additional references are needed [89,90,91,92,93]. It is crucial to understand that these techniques share a commonality in that the cellular samples can originate from primary, metastatic sites or peripheral blood. Subsequently, microfluidic devices or plates are employed to individually capture isolated single cells. The DNA of these captured cells undergoes scWGA, and the resulting genomic DNA libraries are then subjected to high-throughput sequencing. The sequencing data obtained for each individual cell contain valuable information regarding gene expression, mutations, copy number variations, and other genomic features. Single-cell sequencing empowers the identification of distinct cell populations within a tumor, providing a comprehensive view of its heterogeneity. By monitoring genetic changes in individual cancer cells from CTCs, single-cell sequencing can unveil their phenotype and enable the acquisition of crucial characteristics. Through these techniques, the isolation and sequencing of CTCs have revealed the actionability of several molecular alterations, including copy number variations (CNV) and key-driver gene mutations, during the development of resistance in various cancer types, including prostate [94], gastric [95], breast [96], small-cell lung [78], and colorectal cancers [97]. The exceedingly early detection of CTCs with such characteristics could potentially enable the application of highly personalized therapy, targeting a limited number of cells, which would be theoretically more effective than treating a clinically evident disease. These are the frontiers of science where diagnostics and therapy intertwine innovatively, necessitating large-scale clinical studies for validation.

## 6. Conclusions and Perspectives

The presence of CTCs in peripheral blood offers a non-invasive window into the dynamic landscape of cancer progression and treatment response. This review has highlighted their multifaceted role in predicting clinical outcomes and informing prognosis across various solid tumor malignancies. As the field of oncology continues to advance, the transformative impact of CTCs as predictive and prognostic biomarkers in the management of solid tumors will become increasingly evident, offering special promises for targeted and personalized medicine. In the near future, the perspectives surrounding CTCs hold immense promise. One particularly exciting prospect is their potential to serve as indicators of residual disease, especially in the context of oligo-metastatic disease [98]. In fact, the detection and monitoring of CTCs could become a valuable tool in identifying patients at risk of disease recurrence or progression after curative aggressive treatments. This could lead to more targeted and aggressive interventions at an earlier stage, potentially improving the outcomes of oligo-metastatic patients. Furthermore, CTCs have the potential to guide treatment decisions, monitor treatment efficacy, and facilitate precision oncology. As the technologies for CTC detection and isolation continue to improve, their clinical relevance is set to grow.

In the future, research should focus on standardizing CTC detection methods, defining optimal cutoff values for different cancer types, and exploring their role in combination with other biomarkers like circulating tumor DNA (ctDNA) for a more comprehensive assessment of cancer status. Overall, CTCs are poised to play an increasingly crucial role in the diagnosis, prognosis, and treatment of solid tumors, bringing us closer to the era of personalized and precision medicine in oncology.

## Figures and Tables

**Figure 1 cells-12-02590-f001:**
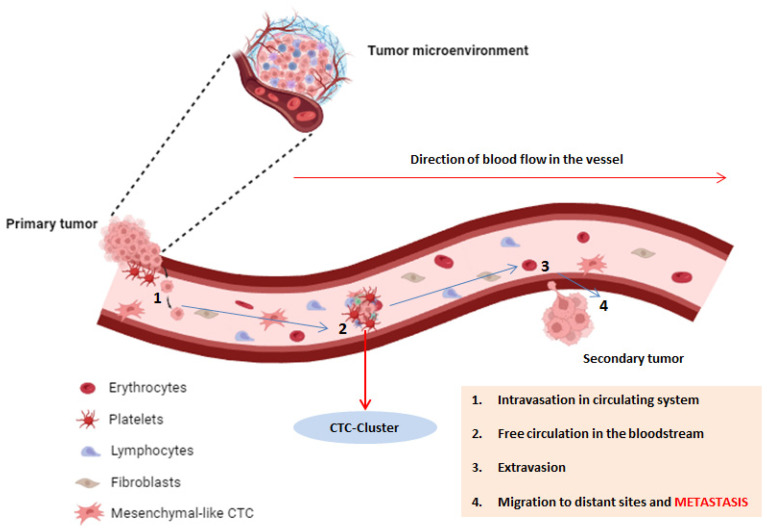
The depiction of CTCs involves their detachment from the primary tumor, intravasation into the bloodstream, and subsequent circulation for the purpose of colonizing distant organs. Following extravasation, they establish secondary metastases. Fundamentally, the process involves circulating tumor cells undergoing epithelial-to-mesenchymal transition (EMT). During EMT, cancerous epithelial cells lose their cell-to-cell contact, adopting a more motile and less differentiated mesenchymal phenotype. CTCs can manifest either as single cells or as cell clusters, the latter exhibiting an enhanced metastatic potential. This diversity in CTC presentation captures a broader spectrum of clonal populations within a tumor, providing a comprehensive portrayal of tumor composition and its dynamic changes over time. Distinguishing themselves from other circulating cells in the blood, CTCs express specific EMT biomarkers, such as epithelial cell adhesion molecules (EpCAM) and cytokeratins (CKs).

**Table 1 cells-12-02590-t001:** Schematic representation of the main CTC label-dependent isolation techniques.

	CTC Isolation Technique	Method	Tumor Type	Clinical Objective	Major Advantages	Major Disadvantages	Ref.
Label-Dependent Isolation	Magnetic nanoparticles	Antibody conjugation to magnetic nanoparticles selectively linked to specific markers expressed by CTCs	Breast Colorectal Prostate Ovarian	Prognosis Treatment	High capture Very Fast	Expensive Low cell viability Low sensitivity	[27]
	Microfluidic chip	Microfluidic chip with functionalized Surfaces in controlled flow to enhance CTC binding to antibody	Breast Prostate Pancreatic Lung	Diagnosis Prognosis Treatment	Cheap High sensitivity High efficiency High cell variability	Small sample vol. Slow flow rate	[28,29]
	Dual modality	Microfluidic platform (columns) with micromagnetic particles functionalized (EpCAM)	Breast Prostate NSCLC Colorectal	Prognosis Diagnosis	Very high efficiency Very high purity	Expensive Medium sensitivity	[30]

Abbreviations: NSCLC: non-small-cell lung cancer; EpCAM: epithelial cell adhesion molecule.

**Table 2 cells-12-02590-t002:** Schematic representation of the main CTC label-independent isolation techniques.

	CTC Isolation Technique	Method	Tumor Type	Clinical Objective	Major Advantages	Major Disadvantages	Ref.
Label-Independent Isolation	Filtration	Filter-based isolation and enrichment	Breast Melanoma Liver Lung Pancreatic	Diagnosis Prognosis Treatment	Very Fast High efficiency	Low purity Filter clogging	[31]
	Microfluidics (no antibodies)	Microscale separation of CTCs according to their dimensions and deformability	Ovarian	Diagnosis	High sensitivity High cell viability	Small sample vol.	[32]
	Density gradient separation	Centrifugal separation of CTCs from blood cells based on their varying densities	Breast Gastric Pancreatic NSCLC	Prognosis	Inexpensive Efficient process	Low purity Loss of cells	[33]
	Imaging	Utilization of a fiber optic array laser scanning system for visual identification of CTCs	Breast Prostate Liver	Diagnosis Prognosis Treatment	High resolution Enumeration of CTCs	Not very precise Difficult sample processing Loss of cells	[34]
Label-Independent Isolation	Dielectrophoresis	Detection of CTCs with application of non-uniform electric field	Breast	Prognosis	High recovery rate High efficiency High cell viability Detection accuracy	Limited volume Low purity Very elaborate procedure	[35]
	Inertial focusing	CTC separation using fluid inertia at high flow rates and inertial focusing	Breast Lung	Diagnosis Prognosis Treatment	High cell viability High precision Speed of processes	Deformation of cells	[36,37]

Abbreviations: NSCLC: non-small-cell lung cancer; CTC: circulating tumor cell.

**Table 3 cells-12-02590-t003:** Selected CTC clinical trials on outcome measures (OS and/or PFS) after the year 2015.

CTC Isolation Technique	Tumor Type	Number of Patients	Outcome Measures	Synthetic Results	Final Remarks	Ref.
Magnetic Nanoparticles (CellSearch)	Metastatic NENs	138	OS	- Alterations in CTCs exhibited a robust correlation with OS (*p* < 0.001) - In multivariate analysis, alterations in CTCs demonstrated the most significant correlation with OS (HR, 4.13; *p* = 0.0002)	- Variations in CTCs exhibit a connection with treatment response and OS in metastatic NENs - CTCs could serve as valuable surrogate markers guiding clinical decision-making	[73]
Magnetic Nanoparticles (CellSearch)	CRC	287	OS and PFS	- CTCs ≥ 1/7.5 mL, showed significant correlation with decreased OS (HR = 5.5) and PFS (HR = 12.7)	- Preoperative CTC detection is a strong and independent prognostic marker in non-metastatic CRC	[74]
Magnetic Nanoparticles (CellSearch)	EC	100	OS	- CTCs detected in 18% of patients - CTCs are robust and independent prognostic indicators for tumor recurrence (HR, 5.063; 95% CI, 2.233–11.480; *p* < 0.001) and OS (HR, 3.128; 95% CI, 1.492–6.559; *p* = 0.003)	- Implementation of CTCs may improve accuracy of preoperative staging in EC	[75]
Magnetic Nanoparticles (CellSearch)	Breast Cancer	3173	OS	- TCs detected in 20.2% of patients - Patients positive for CTCs exhibited larger tumors and more lymph node involvement (*p* < 0.002) - CTCs ≥ 1/7.5 mL were associated with decreased OS (HR: 1.97)	- In patient with primary breast cancer, the existence of CTCs independently forecasted adverse outcomes in terms of disease-free, overall, breast cancer-specific, and distant disease-free survival	[76]
Imaging (Epic Sciences)	mCRPC	161	OS	- The 63% of patients positive for AR-V7-CTC were resistant to ARS inhibitors - Those treated with taxane exhibit a favorable OS compared to those treated with ARS inhibitors (HR = 0.24)	- CTC nuclear AR-V7 protein expression in mCRPC men as a treatment-specific biomarker linked to better survival with taxane therapy than ARS-directed therapy in clinical practice	[77]
Magnetic Nanoparticles (CellSearch)	Lung Cancer	13	OS and PFS	- 88 single CTCs were tested from 13 patients and 83.3% of cases as either chemorefractory or chemosensitive - No significant differences were found in terms of OS, as was found for PFS, between chemorefractory and chemosensitive patients	- The genetic basis for initial chemoresistance differs from that underlying acquired chemoresistance	[78]
Microfluidics (STEMCELL)	CRC	55	OS and PFS	- OS and PFS were 24.2 months and 8.7 months, respectively - CTCs were detected in all the patients - The median OS and PFS were 32.4 and 11.5 months, respectively, in the good prognostic group and 5.4 and 2.7 months, respectively, in the poor prognostic group	- A reliable CTC-based prognostic model has been developed for predicting clinical outcomes in mCRC patients treated with chemotherapy that can be used to help clinicians identify those with the poorest prognosis prior to treatment	[79]
EPISPOT STEMCELL FCM	rHNSCC	65	PFS	- Patients with increasing or stable CTC counts (36/54) from D0 to D7 with EPISPOT^EGFR^ had significantly lower median PFS time (3.9 vs. 6.2 months; 95% CI, 5.0–6.9; *p* = 0.0103); patients with ≥1 CTC detected with EPISPOT or CellSearch^®^ (37/51) had lower median PFS (*p* = 0.0311), as did those with EPISPOT or FCM (38/54) (*p* = 0.0480), and CellSearch or FCM (11/51) (*p* = 0.0005) at D7	- Detection of CTCs is feasible both before and during chemotherapy in individuals with metastatic rHNSCC. The assessment of CTC kinetics from Day 0 to Day 7 using EPISPOT^EGFR^ is linked to treatment response - The potential utility of CTCs as a component of dynamic liquid biopsy for real-time monitoring of early treatment response in patients with rHNSCC undergoing chemotherapy	[80]
EPISPOT CellSearch	CRC	155	OS and PFS	- Patients with persistent viable CTCs between D0 and D28 exhibited shorter PFS (7.36 vs. 9.43 months, *p* = 0.0161) and OS (25.99 vs. 13.83 months, *p* = 0.0178) compared to other patients - The prognostic and predictive values of having ≥3 CTCs, as determined by the CellSearch^®^ system, were confirmed	- The identification of CTCs at Day 28 and the assessment of CTC dynamics from Day 0 to Day 28 using the EPISPOT assay were linked to clinical outcomes and demonstrated potential as predictors of treatment response	[81]

Abbreviations: NENs: neuroendocrine neoplasms; CRC: colorectal cancer; HR: hazard ratio; EC: esophageal cancer; CI: confidence interval; mCRPC: metastatic castration-resistant prostate cancer; AR-V7: androgen receptor splice variant 7; ARS: androgen receptor signaling; rHNSCC: metastatic head and neck squamous cell carcinoma; EGFR: epidermal growth factor; FCM: flow cytometry.

## Data Availability

Non-applicable due to the nature of the study, which is not centered on presenting original data or performing new data analyses.

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
