# Peer review of "Circulating Tumor Cells as Predictive and Prognostic Biomarkers in Solid Tumors"

_cells, 2023, doi:10.3390/cells12222590_

Round 1

Reviewer 1 Report

Comments and Suggestions for Authors

The review focuses on circulating tumor cells, which serve as key biomarkers in theranostics of solid tumors. The importance of their timely detection is also associated with the possibility of non-invasive monitoring of the dynamics of cancer progression during treatment. This is a very dynamically developing field and the authors cite many recent articles, in particular a significant number of publications from 2022-2023.

The manuscript is well-organized and structured, and I have only one suggestion. I missed the author’s view and summary on the issue of the most optimal method of cell isolation. I can advise adding your opinion on this issue.

Author Response

Reviewer #1

The review focuses on circulating tumor cells, which serve as key biomarkers in theranostics of solid tumors. The importance of their timely detection is also associated with the possibility of non-invasive monitoring of the dynamics of cancer progression during treatment. This is a very dynamically developing field and the authors cite many recent articles, in particular a significant number of publications from 2022-2023.

The manuscript is well-organized and structured, and I have only one suggestion. I missed the author’s view and summary on the issue of the most optimal method of cell isolation. I can advise adding your opinion on this issue.

Authors

We would like to express our gratitude to this Reviewer for Her/His appreciation of our work and for granting us the opportunity to share our viewpoint.

We have added this text into the revised manuscript:

“In our opinion, despite the limitations posed by the heterogeneity of marker expression patterns, label-dependent techniques employing antibodies to target CTCs are the most efficient and specific means of CTC isolation. It is also important to note that label-dependent methods are widely used and firmly established in the field. In other words, the biological isolation of CTCs is a highly compelling area of study, and its potential will grow as our knowledge of tumor phenotypes continues to expand. Therefore, the current limitations of label-dependent methods for isolation are more closely related to the constraints of our oncological knowledge than to any inherent "flaw" in the method.”

We have incorporated these perspective elements into the manuscript. The introduction of "In our opinion" is intended to clearly convey to Readers that this represents the authors' viewpoint and may not be universally accepted but is intended to stimulate scientific discourse.

Reviewer 2 Report

Comments and Suggestions for Authors

This review is comprehensive and very well written. The prognostic and predictive value of CTCs are well summarized. However, the great potential to allow the selection of molecular targeted therapies for individualized patient treatments is not presented. Today the most promising agents can be identified also in single cell CTCs based on the associated genes with their mutations, copy number variations and gene expression patterns. Of course, this is a difficult task requiring large clinical studies. The authors should present data form the literature citing these studies already available and in progress.

Author Response

Reviewer #2

This review is comprehensive and very well written. The prognostic and predictive value of CTCs are well summarized. However, the great potential to allow the selection of molecular targeted therapies for individualized patient treatments is not presented. Today the most promising agents can be identified also in single cell CTCs based on the associated genes with their mutations, copy number variations and gene expression patterns. Of course, this is a difficult task requiring large clinical studies. The authors should present data form the literature citing these studies already available and in progress.

Author

We sincerely thank the Reviewer #2 for Her/His appreciation of our work, which provides an up-to-date, and innovative perspective on a such complex topic. Reviewer #2 raises a very elegant and scientifically valuable request. This is a very complex topic; however, in the spirit of a narrative review aimed at providing the necessary information to understand how we are entering a new phase of molecular diagnostics, we have added some reflections in line with what the Reviewer #2 requested.

We have integrated the following text into the manuscript, incorporating specific new references that are mentioned solely within the manuscript:

“The study of circulating tumor cells (CTCs), including novel single-cell techniques, can offer valuable insights into the realm of personalized therapy. In this context, we can reconcile the early characterization of tumor heterogeneity with the resolving power and informative capacity of emerging single-cell technologies. Briefly, single-cell sequencing stands as a groundbreaking approach that allows for the examination of the genetic material of individual tumor cells. This methodology provides insights into the genome, transcriptome, and other multi-omics data at the single-cell level. It's important to note that this section won't delve into the specifics of single-cell whole-genome amplification techniques (scWGA). For detailed information on certain methods such as MALBAC, eMDA, LIANTI, SISSOR, and META-CS, additional references are needed. It is crucial to understand that these techniques share a commonality in that the cellular samples can originate from primary, metastatic sites or peripheral blood. Subsequently, microfluidic devices or plates are employed to individually capture isolated single cells. The DNA of these captured cells undergoes scWGA, and the resulting genomic DNA libraries are then subjected to high-throughput sequencing. The sequencing data obtained for each individual cell contains valuable information regarding gene expression, mutations, copy number variations, and other genomic features. Single-cell sequencing empowers the identification of distinct cell populations within a tumor, providing a comprehensive view of its heterogeneity. By monitoring genetic changes in individual cancer cells from CTCs, single-cell sequencing can unveil their phenotype and enable the acquisition of crucial characteristics. Through these techniques, the isolation and sequencing of CTCs have revealed the actionability of several molecular alterations, including copy number variations (CNV) and key-driver gene mutations, during the development of resistance in various cancer types, including prostate, gastric, breast, small-cell lung, and colorectal cancers. The exceedingly early detection of CTCs with such characteristics could potentially enable the application of highly personalized therapy, targeting a limited number of cells, theoretically more effective than treating a clinically evident disease. These are the frontiers of science where diagnostics and therapy intertwine innovatively, necessitating large-scale clinical studies for validation”.

Reviewer 3 Report

Comments and Suggestions for Authors

Capuozz et al. present an interesting review on the predictive and porgnostic potential of CTCs as biomarkes in solid tumors.

The review covers multiple aspects spanning from characterization of CTS on a mechanistical level to technical aspects of their isolation and clinical implications. While the authors present a very broad overview of the field, the review lacks detail in some aspects.

The statement in line 22 is not justified. The heterogeneity and plasticity of CTCs is not adequately covered in the review and does not give a lot of detail beyond the fact that their are CTTs of mensenchymal- or eptithelial like type.

Line 244 CTCs typically lack the CD45...

While it is true that CD45 is typically only found on leukocytes (not on CTCs), CTCs do not always express EpCAM and CK

line 278: EpCAM-dependent device

The "device" should be explicityl mentioned

line 306

The statement is not very clear to me. Also only on reference is given, which focuses on targeting CTCs in immunotherapy. Are their also other ongoing retrsopective studies?

There is a period missing in line 172.

Comments on the Quality of English Language

The review is well written. Some typos might need corrections.

Author Response

Reviewer #3

Capuozz et al. present an interesting review on the predictive and porgnostic potential of CTCs as biomarkes in solid tumors.

The review covers multiple aspects spanning from characterization of CTS on a mechanistical level to technical aspects of their isolation and clinical implications. While the authors present a very broad overview of the field, the review lacks detail in some aspects.

The statement in line 22 is not justified. The heterogeneity and plasticity of CTCs is not adequately covered in the review and does not give a lot of detail beyond the fact that their are CTTs of mensenchymal- or eptithelial like type.

Authors

To accomplish Reviewer #3 criticism and be more consistent with review contents we have rephrased line 22 as follows: “Moreover, we have underscored the dynamic nature of CTCs and their implications for personalized medicine.”

Reviewer #3

Line 244 CTCs typically lack the CD45... While it is true that CD45 is typically only found on leukocytes (not on CTCs), CTCs do not always express EpCAM and CK

Authors

We concur with Reviewer #3. Her/His critique is not only constructive but also an opportunity to enhance the scientific accuracy of our work. This critical feedback enriches our research. In response to Reviewer #3's request, we have integrated the following text into the manuscript, incorporating specific new references that are mentioned solely within the manuscript:

“As previously elucidated, the majority of cancers originate from epithelial cells, making EpCAM the most commonly used marker for CTCs. EpCAM is considered a "universal" epithelial marker for various cancer types. However, the utility of EpCAM as a CTC marker is constrained. It cannot be applied to tumors that lack EpCAM expression or have low levels of expression, such as neurogenic cancers. Additionally, CTCs can undergo epithelial-to-mesenchymal transition (EMT), during which epithelial markers like EpCAM are downregulated. This phenomenon hampers the detection of EpCAM-positive CTCs. Hence, it is noteworthy that relying solely on the detection of EpCAM-positive CTCs likely underestimates the total CTC population and overlooks crucial biological information pertaining to EpCAM-negative CTCs, i.e., those undergoing EMT. From a methodological perspective, the challenge is compounded by the fact that many EMT-related molecules are cytoplasmic or nuclear proteins, rendering them unsuitable for use with currently available membrane-based molecular technologies for CTC detection. Therefore, single-cell CTC sequencing technologies (see beyond) may prove more beneficial for enhancing our understanding of the phenotype of EpCAM-negative CTCs and improving their isolation, enabling a more comprehensive assessment of the EMT status of CTCs at the RNA level".

Reviewer #3

line 278: EpCAM-dependent device. The "device" should be explicityl mentioned.

Authors:

The device has been detailed. Please read the revised version.

Reviewer #3

line 306

The statement is not very clear to me. Also only on reference is given, which focuses on targeting CTCs in immunotherapy. Are their also other ongoing retrsopective studies?

Authors

The Reviewer #3 is entirely correct, and we apologize for the error. The passage has been entirely rephrased. Thank you. “Moreover, the identification of new markers poses technical challenges [70]. Technological advancements and extensive clinical validation are essential to understand CTCs role in guiding personalized medicine and drug development.”

Reviewer #3

There is a period missing in line 172.

Authors

We apologize; it has been rectified. It was due to issues with the MDPI template.